# Raman Spectroscopy Spectral Fingerprints of Biomarkers of Traumatic Brain Injury

**DOI:** 10.3390/cells12222589

**Published:** 2023-11-08

**Authors:** Georgia Harris, Clarissa A. Stickland, Matthias Lim, Pola Goldberg Oppenheimer

**Affiliations:** 1Advanced Nanomaterials Structures and Applications Laboratories, School of Chemical Engineering, College of Engineering and Physical Sciences, University of Birmingham, Edgbaston, Birmingham B15 2TT, UK; 2Institute of Healthcare Technologies, Mindelsohn Way, Birmingham B15 2TH, UK

**Keywords:** traumatic brain injury, TBI biomarkers, acute, sub-acute and chronic phases, Raman spectroscopy, neurodiagnostics

## Abstract

Traumatic brain injury (TBI) affects millions of people of all ages around the globe. TBI is notoriously hard to diagnose at the point of care, resulting in incorrect patient management, avoidable death and disability, long-term neurodegenerative complications, and increased costs. It is vital to develop timely, alternative diagnostics for TBI to assist triage and clinical decision-making, complementary to current techniques such as neuroimaging and cognitive assessment. These could deliver rapid, quantitative TBI detection, by obtaining information on biochemical changes from patient’s biofluids. If available, this would reduce mis-triage, save healthcare providers costs (both over- and under-triage are expensive) and improve outcomes by guiding early management. Herein, we utilize Raman spectroscopy-based detection to profile a panel of 18 raw (human, animal, and synthetically derived) TBI-indicative biomarkers (N-acetyl-aspartic acid (NAA), Ganglioside, Glutathione (GSH), Neuron Specific Enolase (NSE), Glial Fibrillary Acidic Protein (GFAP), Ubiquitin C-terminal Hydrolase L1 (UCHL1), Cholesterol, D-Serine, Sphingomyelin, Sulfatides, Cardiolipin, Interleukin-6 (IL-6), S100B, Galactocerebroside, Beta-D-(+)-Glucose, Myo-Inositol, Interleukin-18 (IL-18), Neurofilament Light Chain (NFL)) and their aqueous solution. The subsequently derived unique spectral reference library, exploiting four excitation lasers of 514, 633, 785, and 830 nm, will aid the development of rapid, non-destructive, and label-free spectroscopy-based neuro-diagnostic technologies. These biomolecules, released during cellular damage, provide additional means of diagnosing TBI and assessing the severity of injury. The spectroscopic temporal profiles of the studied biofluid neuro-markers are classed according to their acute, sub-acute, and chronic temporal injury phases and we have further generated detailed peak assignment tables for each brain-specific biomolecule within each injury phase. The intensity ratios of significant peaks, yielding the combined unique spectroscopic barcode for each brain-injury marker, are compared to assess variance between lasers, with the smallest variance found for UCHL1 (*σ*^2^ = 0.000164) and the highest for sulfatide (*σ*^2^ = 0.158). Overall, this work paves the way for defining and setting the most appropriate diagnostic time window for detection following brain injury. Further rapid and specific detection of these biomarkers, from easily accessible biofluids, would not only enable the triage of TBI, predict outcomes, indicate the progress of recovery, and save healthcare providers costs, but also cement the potential of Raman-based spectroscopy as a powerful tool for neurodiagnostics.

## 1. Introduction

Traumatic brain injury (TBI) is the leading cause of death under the age of 40, presenting a major burden on healthcare services worldwide [1]. The severity of TBIs ranges from mild/moderate concussion to severe and chronic loss of cognitive and motor function, resulting from sudden impact to the head, often following road accidents, falls, assaults, and in contact sports [2]. The physiological symptoms of TBI vary from fatigue and headaches in mild injuries, to temporary (and occasionally permanent) loss of memory or consciousness, short and long-term disability, neurological disorders and death, in the more severe scenarios [3,4]. Importantly, recent research [3,4,5,6,7,8] has been focusing on establishing the temporal profile of TBIs and the associated biomarkers including their emergence (i.e., acute phase), crucial for early-stage diagnostics, persistence, and decline (sub-acute and chronic phases)—indicative of the longer-term effects and prognosis as well as development of new therapeutics. TBI biomarkers have been shown to have a characteristic timeframe of acute (<24 h), sub-acute (1 day–3 weeks) and chronic (3 weeks–6 months), as the initial insult causes acute damage to the brain, is typically followed by a biochemical cascade leading to secondary injuries over the following month [9,10,11]. This indicates that identifying TBI in the acute phase and intervening before further damage occurs is vital for long-term neurological recovery. However, despite the important time window alongside an estimate of 69 million people suffering from TBI each year worldwide, the development of point-of-care, timely TBI diagnostic technologies remains an unmet need and there is no definite triaging and therapeutic patient pathway [12].

Current TBI diagnostic techniques include the Glasgow Coma Scale (GCS), a cognitive assessment method, and neuroimaging such as Computerised Tomography (CT) and Magnetic Resonance Imaging (MRI) scans [2]. The GCS uses the patient’s eye, verbal, and motor responses to enumerate the injury and categorise it as either severe (3–8), moderate (9–12), or mild (13–15) [13]. Whilst neuroimaging will identify skull fractures, hematoma, and haemorrhage to determine if neurosurgery is necessary [2]. The main challenge lies in the fact that pre-hospital and emergency department (ED) assessment is predominantly based on insensitive GCS and history acquisition with guidance on CT being often vague and costly. Mild TBI (mTBI) is often underdiagnosed if not identified rapidly at the point of injury, with 90% of acute cases undetected and not admitted to ED [2,14,15]. Therefore, development of successful TBI biomarker diagnostics from blood [16], or other biofluids, i.e., cerebrospinal fluid (CSF) [8], saliva [17], urine [18], and tears [19], would lead to improved triage of severe and high-risk injuries and improve outcomes, whilst appropriate classification of low-risk and mTBI patients would allow for the focusing of resources on those who need it the most. Furthermore, reduced resource demand would mitigate the relatively low costs of the investigation in comparison to ED assessment, CT imaging and hospital admission, and have distinct yet complementary use cases. In the pre-hospital space, improved triage of TBI as mild will support decision-making by pre-hospital healthcare providers to reduce referral to hospital care, reducing the number of ED visits.

Concurrently, Raman spectroscopy, a rapid, label-free technique has been increasingly used in medical and diagnostic applications for producing non-destructive chemical information [20]. This sensitive vibrational spectroscopy excites molecular bonds within a sample, providing a unique biomolecular spectral fingerprint of target biomarkers in a rapid analytical response [21,22]. These spectroscopic barcodes have been shown to identify the various diseases from which the biofluid has been collected, such as blood [23,24], CSF [25,26], urine [27,28], saliva [24,29], and tears [30,31]. In contrast to the in vitro bioassays, the availability of inexpensive, portable Raman devices makes this technique particularly attractive for point-of-care testing, analysis, and screening of biofluids. There is limited yet growing research in the development of Raman spectroscopy for biofluid-based neurodiagnostics [32,33,34,35,36]. Whilst research continues to unravel new technologies for rapid point-of-care technologies for TBI diagnostics from biofluids and tissue, it is imperative to establish fingerprints of the inherent characteristics of TBI biomarkers in various injury phases, where the detected changes via Raman spectroscopy could, in the long-term, be attributed to underpinning the variations in TBIs with various severity and as a function of temporal evolution. 

Herein, we consistently study a broad panel of TBI biomarkers via Raman spectroscopy profiling and establish a spectral reference library for the interpretation of the Raman fingerprints of TBI-indicative biomarkers via biologically applicable laser excitation wavelengths in different post-injury phases. A carefully selected cohort of TBI-indicative biomarkers are classed into three main injury phases with spectra in the raw and reconstituted (mimicking the biomarker biofluid) state forms acquired via a range of Raman-applicable laser excitation wavelengths of 514, 633, 785, and 830 nm. Subsequently, for each post-TBI phase, Raman spectroscopy temporal-phase biomarker-profiling establishes an important baseline library in the form of a “multi-biochemical barcode”, yielding a characteristic tool for ongoing and future spectroscopic studies for diagnostic applications. Awareness of the spectroscopic temporal profiles of TBI biomarkers provides an essential pathway for defining and setting the most appropriate diagnostic time window for sampling after injury. The derived Raman fingerprints, combined with the identified biochemical peaks, closely matching notable biomarker characteristics, equip us with the knowledge of the molecule’s physiological role. This could not only improve the treatment of TBI through specific targeting of the damage in contrast to current methods, which mostly rely on symptomatic relief [37], but could also provide a further panel of candidate first-line screening TBI-indicative biomarkers. This lays the platform for defining their functionality in the complex TBI aetiology and, in the longer term, cements Raman spectroscopy as a powerful technique for future biomarker discovery in both neurodiagnostics as well as for other detrimental diseases with many ramifications.

## 2. Materials and Methods

### 2.1. TBI Biomarkers

The selected raw biomarkers were purchased without purification and tested in both solid and solution states. These were divided into three main groups of acute, sub-acute, and chronic phases (Figure 1). The TBI cohort selection was based on an extensive literature overview of recent research on TBI biomarkers (animal and human) and temporal courses studied in a range of biofluids including blood, plasma, CSF, urine, and the key biomarkers of neuroinflammation [3,38,39,40,41,42,43,44,45]. Biomarkers included in the panel were obtained in the purest form without any active additives, to avoid impacting spectral signatures. This yielded a cohort of 18 biomarkers including the N-Acetyl-L-aspartic acid (NAA), ganglioside, Glutathione (GSH), Neuron-Specific Enolase (NSE), Glial fibrillary acidic protein (GFAP), Ubiquitin carboxyl-terminal hydrolase isozyme L1 (UCHL1), cholesterol, D-serine, sphingomyelin, sulfatide, cardiolipin, Interleukin-6 (IL-6), S100B, galactocerebroside, glucose, myo-inositol, Interleukin-18 (IL-18), and Neurofilament light chain (NFL). Subsequently, reconstituted biomarker samples were prepared following the technical details provided in the individual data sheets. Typically, this included centrifugation at 8 G for 2 min prior to adding the solvent of distilled H_2_O (18.2 MΩ) or chloroform/ethanol, yielding 1:1 *v*/*v* solutions. After 20 min, to allow full dissolvement, 3 μL samples of each biomarker were deposited onto an aluminium slide and dried in an ambient environment for Raman measurement. 

### 2.2. Reference Chemicals

Human derived biomarkers included the S100B (HY-P70659, Cambridge Bioscience, Cambridge, UK), IL-6 (H7416-10UG, Merck, Darmstadt, Germany), NSE (13219-H08E-SIB-50 ug, Stratech, Ely, UK), GFAP (C227-10 ug, Generon, Slough, UK), IL-18 (CSB-YP614514HU, Antibodies.com, Cambridge, UK), NFL (pro-2584-10 ug, Generon), and UCHL1 (UC1-H5140-50 ug, Generon). Animal derived biomarkers included the galactocerebroside (C4905-10MG, Merck), sphingomyelin (1051-25 mg, Cambridge Bioscience), ganglioside (860053P-10MG, Scientific Laboratory Supplies, Nottingham, UK), sulfatide (56-1085-7-LAO-25 mg, Stratech), and cardiolipin (C0563-10MG, Merck). Synthetically produced biomarkers included the NAA (00920-5G, Merck), cholesterol (C8667-1G, Generon), glucose (CAY23733-50, Cambridge Bioscience), D-serine (A11353.06, VWR International, Lutterworth, UK), myo-inositol (A13586.22, VWR International), and GSH (G4251-300MG, Merck). 

**Figure 1 cells-12-02589-f001:**
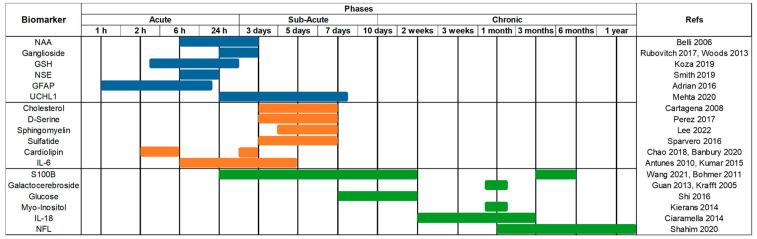
Schematic timeline overview of the TBI biomarkers and their phases, acute (blue), sub-acute (orange), and chronic (green), during which each biomarker has been reported to either increase, persist, or decline post-injury [4,9,40,44,46,47,48,49,50,51,52,53,54,55,56,57,58,59,60,61,62,63,64]. Biomarkers: N-Acetyl-L-aspartic acid (NAA), Glutathione (GSH), Neuron-Specific Enolase (NSE), Glial fibrillary acidic protein (GFAP), Ubiquitin carboxyl-terminal hydrolase isozyme L1 (UCHL1), Interleukin-6 (IL-6), Interleukin-18 (IL-18), and Neurofilament light chain (NFL).

### 2.3. Raman Spectroscopy

Raman measurements were carried out using the InVia Qontor confocal Raman system (Renishaw) equipped with four excitation lasers, which were adjusted for optimal throughput, fluorescence control, and sensitivity. Optical measurements were carried out with a specially adapted research-grade microscope (Leica [Wetzlar, Germany] DM 2700M) equipped with an incoherent white light source, allowing confocal measurements with a 2.0 μm depth resolution. Four excitation wavelengths of 514, 633, 785, and 830 nm were employed in this study to enable a breadth of spectral profiles. To avoid photochemical effects in the spectra, sample damage or degradation, extended spectral Raman scans were acquired over a range of 200–3200 cm^−1^, 50× objective lens, and an acquisition time of 10 s with 5 accumulations. All Raman spectra were collected at the ambient temperature. 

### 2.4. Data Analysis

The acquired Raman spectra were collected using Renishaw WiRE 5.2 (Renishaw Plc, Wotton-under-Edge, UK) and cosmic rays were removed during baseline subtraction with a polynomial order of 11 and a 1.5 noise tolerance. Extended spectra were subsequently averaged per biomarker, taking into account the laser excitation wavelength as well as the raw and reconstituted states. The Peak Pick tool (WiRE 5.2) was utilised across the x-axis range to select the six characteristic bands for each assignment, while the slope detection method was utilised with 15 smooth points and a peak threshold height of 500 counts.

## 3. Results and Discussion

The Raman spectrum is commonly referred to in terms of discrete regions, the low wavenumber region (100–200 cm^−1^), the fingerprint region (500–2000 cm^−1^), and the high wavenumber region (2000–4000 cm^−1^) [65,66]. In this study, Raman spectra were collected in the extended spectral regions of 200–3200 cm^−1^, however, only the fingerprint region from 200 to 1800 cm^−1^ was used where the main bands of interest were identified. 

The biomarkers studied, based on references to TBIs in the literature, were classified within three main injury phases as acute, sub-acute, and chronic (Figure 1), the corresponding detailed information for each biomarker, including the accessible biofluid source, physiological function, post-TBI response, and the role they play in head injuries along with the associated phase behaviour are summarised in Table 1.

Spectral profiles for each biomarker were acquired for both solids as well as of the solute forms, with the former inherently yielding better resolved, sharper Raman peaks. It is well-established that Raman band frequencies are more complex and shifted in the liquid phase, relative to the solid, due to the higher variability of the intermolecular forces and molecular collisions in the liquid phase, which shift the frequencies of intra-molecular vibrations and broaden the bands, relative to the highly regular and stable solid structures. Since the behaviour of biomarkers in aqueous solutions is of particular importance, given that an aqueous medium is a natural environment for biological molecules, we have identified the spectral signatures of the reconstituted solutions of TBI biomarkers classified by injury state (Figure 2, Figure 3 and Figure 4). Further studied variation, due to the excitation wavelengths along with the comparison between the raw and reconstituted biomarkers’ form, are shown in Appendix A.

**Table 1 cells-12-02589-t001:** Overview of the studied TBI biomarker cohort with the corresponding physiological significance. Source refers to the biofluid in which the biomarker has been analysed in the literature. N-Acetyl-L-aspartic acid (NAA); Glutathione (GSH); Neuron-Specific Enolase (NSE); Glial fibrillary acidic protein (GFAP); Ubiquitin carboxyl-terminal hydrolase isozyme L1 (UCHL1); Inter-leukin-6 (IL-6); Interleukin-18 (IL-18); and Neurofilament light chain (NFL). Increase in concentration (↑), decrease in concentration (↓).

Biomarker	Source	Physiological Function	TBI Role	TBI Response	Reference
NAA	Blood CSF	Synthesised in neurons, specific to the nervous system. Has roles in maintaining myelin lipid synthesis and in promoting neuronal mitochondria ATP production.	Marks injury type. Depletion in grey matter and white matter represent neuronal loss and axonal damage, respectively. Rate of replenishment is inversely proportional to injury severity.	↓	[33,67,68]
Ganglioside	Serum CSF Brain	Cellular signalling, protein, and ion channel modulator. Main carrier of sialic acid in the nervous system, improving intercellular communications.	Functional disruption causes neurodegeneration, cellular dysfunctions and promotes disease pathogenesis.	↑	[69]
GSH	Blood CSF	Essential antioxidant, converted from its reduced state (GSH) to its oxidised form to regulate free radicals from the brain.	GSH depletion coupled with neuroinflammation leads to build-up of free radicals, further damaging the brain and neurons.	↓	[61,70]
NSE	CSF	Tissue-specific cytosol-based enzymes, upon stimulation they translocate to the cell surface to act as a plasminogen receptors.	The degree of damage is directly correlated to the amount of NSE expressed.	↑	[71]
GFAP	Blood CSF	Intermediate filament III protein. Maintains glial cytoskeleton structure, neighbouring neurons, and blood-brain barrier (BBB).	Post-trauma, astroglial cells undergo astrogliosis, causing cellular hypertrophy and increased GFAP expression. Excess GFAP can cause glial scars in brain tissue which delays axon regeneration.	↑	[72,73,74]
UCHL1	Blood CSF	A brain-specific enzyme features in the ubiquitin-proteasome pathway to maintain axonal protein integrity. Regulates axonal transport, structure, and synapsis.	Selectively increases upon axonal damage, utilised to remove damaged and defective proteins from axons.	↑	[75,76]
Cholesterol	CSF Brain Serum	Key component of the cellular membrane, maintaining structure and fluidity. Cholesterol and phospholipids are transported to nerve cells for repair, upkeep and to promote neurite proliferation.	Increased cholesterol is in proportion to cellular damage. Removing excess cholesterol has an anti-inflammatory effect. Dysregulation of brain cholesterol negatively affects neuronal and glial function. Cholesterol builds up from dysregulation causes cellular toxicity.	↑	[64,77,78,79,80]
D-Serine	Brain CSF Blood	*α*-amino acid, abundant in the brain. Binds to N-methyl-d-aspartate (NMDA) and δ_2_ glutamate receptors to contribute to learning and memory function.	Reactive glial cells become D-Serine synthesisers under inflammatory conditions, causing NMDA receptor hyperactivation, and leading to hippocampal synaptic damage.	↑	[81]
Sphingomyelin	CSF Blood	Vital in regulation of cellular growth rate, differentiation, and death in the central nervous system. Supports myelination in the brain, and aids in cognitive maturation and regulation of inflammatory responses.	Increased levels are reported in the hippocampus over 12 months after TBI, contributing to neurological disease pathogenesis. Breakdown products regulate the sphingomyelin cycle which inhibits protein kinase c, regulating neuronal signal transduction and function.	↑	[82,83,84,85,86]
Sulfatides	Brain CSF Serum	Abundant in myelin sheath and myelinating cells. Negatively regulates and improves oligodendrocyte differentiation and survival. Maintains myelin and axonal-glial signalling.	Effects functional properties of the membrane, and dysregulation of sulfatides can lead to seizures.	↓	[87]
Cardiolipin	Inner Mitochondrial Membrane	Involved in regulating mitochondrial metabolism. The structural component of mitochondrial membranes regulates protein and enzyme activity central to mitochondrial function.	Damaged mitochondria trigger neuronal death when oxidised. Cardiolipin acts as an elimination signal for damaged mitochondria, thus limiting neuronal damage and preserving cognitive functions.	↑	[88,89,90]
IL-6	CSF Serum Blood	Pleiotropic cytokine with operations in immunity regulation, regeneration processes, neural functions, and cardiovascular protective mechanisms.	Usually undetectable in healthy brain parenchyma but present within an hour following TBI. Upregulated production following trauma from inflammatory cascades to salvage neurons. Sustained inflammation is ultimately damaging.	↑	[52,91,92,93]
S100B	Blood CSF Brain	Calcium-binding protein involved in long-term synaptic plasticity modulation, cellular growth and structure, calcium concentration maintenance, and energy metabolism. Mitigates mitochondrial failure through calcium modulation.	Overexpression leads to disrupted calcium homeostasis. Increased levels indicate structural damage and cellular death.	↑	[94,95]
Galactocerebroside	CSF Brain	Major lipid component in the brain, which maintains myelin sheath structure and stability. Important for development of normal myelin in the central nervous system.	Galactosylcerimidase dysfunction prevents Galactocerebroside from degrading, instead it accumulates in globoid cells in the brain, leading to white matter diseases.	↑	[55,96,97]
Glucose	Blood	Main energy source to the brain, used for action potential and postsynaptic potential generation. Synthesises BBB-regulated neuroactive compounds and sustains brain homeostasis.	Early low glucose levels and low lactate/glucose ratio post-TBI are associated with poor outcomes. Not associated with ischemia.	↓	[98,99,100]
Myo-Inositol	CSF Blood	Regulates glial and neuronal activity and participates in intracellular signalling pathways. Regulates intracellular [Ca^2+^] and membrane permeability.	Increased levels are correlated to glial proliferation, increased rate of membrane turnover, and myelin sheath damage. Correlated to astrogliosis and dysregulation of cellular osmotic functions.	↑	[101,102,103,104]
IL-18	CSF	Inducer of inflammatory cytokines; synthesised as an inactive precursor in microglia and activated by caspase-1 in a forward loop.	Has roles in neuroinflammation and neurodegeneration. Induces respiratory burst and degranulation of polymorphonuclear leukocytes resulting in a release of neurotoxic enzymes.	↑	[105,106]
NFL	CSF Blood	Integral in maintaining axonal cytoskeleton through radial growth. Larger myelinated axons result in more NFL, leading to faster conduction speed.	NFL levels rapidly increase following trauma to account for damaged axons and NFL is released into the interstitial space and integrated into CSF. Indicates significant axonal damage and progression rate of disease.	↑	[107,108,109]

### 3.1. Acute Phase

Acute TBI biomarkers have been shown to be applicable in triaging, diagnosing, and eliminating the presence of TBI at the earliest stages (i.e., golden-hour biomarkers) and are thus particularly important for the development of diagnostic point-of-care modalities as well as intervention for TBI [110,111]. Therefore, biochemical changes and concentration variations in biomarkers are reflected through the spectral information when comparing healthy control cohorts to acute, sub-acute, and chronic phases, which are vital for PoC diagnostics and injury monitoring. Representative average spectra of acute TBI biomarkers are presented in Figure 2 alongside peak assignment provided in Table 2. Each biomarker produced unique and distinctive spectral features. 

Strong characteristic peaks of NAA are found in the 949–957 cm^−1^ and 987–992 cm^−1^ regions for the four-excitation laser used (Figure 2a), with the main peaks being associated with the C-N stretching as well as CH_2_-CH wagging [112]. NAA is an amino acid derivative located in neurons where the decreasing levels are proportional to neuronal loss and axonal damage [68] and, thus, a decrease in characteristic Raman peak intensity can be an important indicator following brain trauma. 

The most intense peaks of ganglioside, a sialic acid-bearing glycosphingolipid [60], are detected at 1297–1298 cm^−1^ and 1437–1442 cm^−1^ (Figure 2b). The bands at 1297 and 1440 cm^−1^ are assigned to bending and twisting of CH_2_ bonds [113], which optimally could be employed relative to other characteristic peaks of ganglioside post-TBI [114]. These peak intensities have been reported to decrease in counts in tandem with cell loss [114], in contrast to the other characteristic peaks of ganglioside (Table 2), which are typically shown to increase in response to neurodegeneration and cellular dysfunction (Table 1) [69].

GSH, a non-enzymatic antioxidant tripeptide present in cells to protect membranes from oxidative damage [115], exhibits the most intense peaks located at 1420–1424 cm^−1^ and 1660–1662 cm^−1^ (Figure 2c), attributed to the -CH_2_ bending mode of proteins [116], and symmetrical stretching of the carboxylic acid COO^−^ [117], and the Amide I region, indicative of amino acids [118], respectively. Post-TBI apoptosis could lead to GSH depletion and thus a reduction in characteristic peak intensity which, when coupled with neuroinflammation, results in TBI secondary injury due to free radicals build-up further damaging neurons and the brain (Table 1) [70].

NSE is a glycolytic tissue-specific enzyme associated with neuronal damage and post-traumatic inflammation [41,63], increasing in expression with injury severity [71]. The strongest peak for NSE (Figure 2d) at 875–877 cm^−1^ at all four employed excitation wavelengths is assigned to the stretching of C-C bonds and the asymmetric stretching of C-N bonds [55,119] associated with phenylalanine. A secondary peak of interest at 1077–1082 cm^−1^ is attributed to the stretching of C-O and C-C bonds [120] along with the twisting of CH_2_ bonds [121]. NSE is plausible for monitoring post-TBI changes since it is only expressed in a direct correlation to the degree of damage and translocated from the cytosol to cell surface upon stimulation, acting as a plasminogen receptor [62,71,122].

**Figure 2 cells-12-02589-f002:**
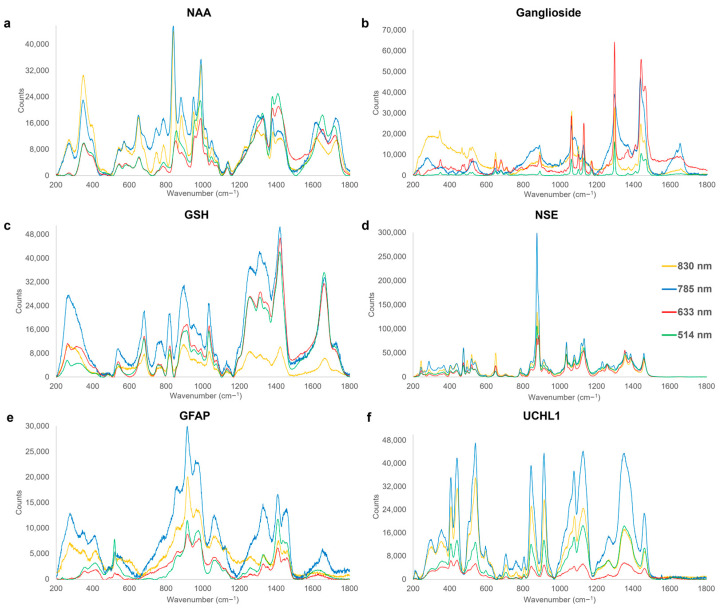
Characteristic Raman spectra fingerprints of acute-phase TBI biomarkers, acquired at excitation wavelengths of 514 nm (green), 633 nm (red), 785 nm (blue), and 830 nm (yellow). Significant Raman peak assignments for each are summarized in Table 2. N-Acetyl-L-aspartic acid (NAA); Glutathione (GSH); Neuron-Specific Enolase (NSE); Glial fibrillary acidic protein (GFAP); and Ubiquitin carboxyl-terminal hydrolase isozyme L1 (UCHL1).

GFAP is a structural filament protein of astrocytes [15], with its levels known to increase in response to astrogliosis post-trauma, as discussed in Table 1. The most intense detected spectral bands are found in the region of 914–918 cm^−1^ (Figure 2e), assigned to C-N stretching [119], and asymmetric vibration of the pyranose ring [112], also present in the UCHL1 spectrum (Figure 2f), increasing in response to TBI, and thus might collectively be an important candidate peak of interest for TBI Raman-based diagnostics.

A further sharp peak present in the 1408–1410 cm^−1^ region is assigned to the bending of C-H bonds and symmetrical stretching of COO^−^ bonds [119,123], along with a wider band at 1060–1068 cm^−1^, assigned to the skeletal stretching of C-C bonds and indicative of changes in protein levels [124]. GFAP has been shown to increase after brain trauma because of reactive astrogliosis. Its overexpression, causing scarring of the brain, can be used to distinguish between the healthy and damaged tissue, giving an indication of the amount of damage, and delaying the axonal regeneration [73].

**Table 2 cells-12-02589-t002:** Acute TBI phase associated Raman bands and their assignments. *v* = stretching; *δ* = bending; *τ* = twisting; *ρ* = rocking; *ω* = wagging; *s* = symmetric; *a* = anti-symmetric; *arom* = aromatic; *skel* = skeletal.

Wavenumber (cm^−1^)	Assignment	Origin	Reference
347–353	*v_skel_*(C-C)	NAA	[112]
473–475	*v*(S-S)	NSE	[125]
518–521	C(OH), *Ring deformation*	GFAP	[112,119]
647–651	*v*(C-S), *τ*(C-C), *Ring breathing mode*	NAA, NSE, Ganglioside	[119,126,127]
679–682	*v*(C-S)	GSH	[119]
705–708	*Ring deformation*	UCHL1	[119]
806	*v_s_*(C-N-C)	UCHL1	[112]
819–821	CH *deformation*	GSH	[112]
843–845	*δ*(H(C-O-H)), *ρ*(H(C-O-H))	UCHL1	[119]
875–877	*v*(C-C), *v_a_*(C-N)	NSE	[55,119]
889–891	*v_arom_*(C-O), *v*(C-C), *ω*(CH_2_)	Ganglioside	[128,129,130]
914–918	*v*(C-N), *Pyranose ring asymmetric vibration*	GFAP, UCHL1	[112,119]
949–957	*v*(C-N)	NAA	[112]
974–980	*v_a_*(C-C), *ρ*(CH_2_)	GFAP	[119,131]
987–992	*v*(C-N), *ω*(CH_2_- CH)	NAA	[112]
1031–1037	*δ*(C-H), *v*(C-C), *v_a_*(C-C-N^+^)	GSH, NSE	[132,133]
1060–1068	*v*(C-O), *v_skel_*(C-C), *τ*(NH_2_)	Ganglioside, GFAP	[119,124,134]
1077–1082	*v*(C-O), *v*(C-C), *τ*(CH_2_)	NSE, UCHL1	[120,121]
1127–1130	*v*(C-N), *v*(C-C)	Ganglioside, UCHL1	[135,136]
1131–1138	*v*(C-C)	NAA	[137]
1297–1298	*τ*(CH_2_)	Ganglioside	[114]
1311–1315	*δ*(C-H)	GSH	[138]
1328–1336	*ω*(CH_2_), *τ*(CH_2_)	GFAP	[119]
1352–1354	*τ*(CH_2_)	UCHL1	[121]
1378–1383	*v_s_*(COO^−^)	NAA	[139]
1408–1410	*δ*(C-H), *v_s_*(COO^−^)	GFAP	[119,123]
1420–1424	*δ*(CH_2_), *v_s_*(COO^−^), *v_s_*(CO_2_)	GSH	[116,117,138]
1437–1442	*δ*(CH_2_), *τ*(CH_2_)	Ganglioside	[113]
1458–1460	*δ*(CH_2_)	NSE	[140]
1660–1662	*v*(C-N(H)-C=O), *v*(C=C)	GSH	[112]

UCHL1, a cytoplasmic enzyme found in neurons, is selectively increased following axonal trauma to remove damaged and defective proteins from axons [75,141]. Unfortunately, the glycerol-environment buffer of this protein dominates the spectral fingerprint with aliphatic chains [121,142] and, therefore, only smaller peaks in the 600–800 cm^−1^ region are identified to be unique to the UCHL1 marker (Figure 2f) with the 705–708 cm^−1^ band assigned to ring deformation [119], and the 806 cm^−1^ peak assigned to the symmetrical stretching of C-N-C bonds present in amino acids [112].

### 3.2. Sub-Acute Phase

Detection of sub-acute TBI biomarkers is vital for multiple-level diagnostics when the patient is admitted to hospital and a complementary rapid diagnostic technique is required alongside structural neuroimaging. This lays the platform for not only enabling timely TBI management at ED, A&E, etc. for early neuroprotective measures but also for the correct transfer to the most appropriate neurological facility. Rapid diagnosis in the early clinical phase will lay a platform for a range of improvements in personalised medicine and management, reduce strain on the healthcare system, and enable better-quality post-neurotraumatic care. Similarly, in the military context, where neurosurgical support is not routinely deployed and TBI management often requires evacuation, the ability to diagnose and assess TBI severity pre-hospital would avoid unnecessary strategic evacuation and maintain operational effectiveness. Representative average spectra of sub-acute TBI solute-form biomarkers are shown in Figure 3 with the corresponding peak assignments summarised in Table 3. 

Cholesterol’s sterol-centred structure presents a unique and large number of intense peaks (Figure 3a), including the 700 cm^−1^ band of the in-plane deformation of its B ring [143], as well as a sharp peak at 1673 cm^−1^, assigned to the B ring C=C stretching [144]. Cholesterol is structurally important in the cellular membrane with its increases following TBI proportionally related to the cellular damage [145]. As a response to neuro-inflammation, apolipoprotein E transports cholesterol and phospholipids by forming a protein–lipid complex to damaged nerve cells to assist the cellular damage as well as to increase neurite proliferation [77,78]; whilst cholesterol removal is associated with the anti-inflammatory response [64,79]. If left unchecked, the dysregulation of cholesterol in the brain can negatively affect neuronal and glial functions and in due course, leading to cellular toxicity [80].

D-Serine is a non-essential polar amino acid, which serves as a precursor to purines and pyrimidines, and is an important neurotransmitter [146]. The characteristic fingerprint of D-Serine (Figure 3b) exhibits strong peaks in the range of 917–923 cm^−1^, attributed to its C-C backbone stretching [140], as well as at 1327–1336 cm^−1^ from the twisting of its CH_2_ groups [119]. Serine is a co-ligand activator of NMDA receptors and of δ_2_ glutamate receptors, which have regulatory roles in synaptic plasticity and long-term potentiation [81]. Increases in serine following TBI is caused by reactive glial cells, which under inflammatory conditions turn into D-serine synthesisers. An increase in D-serine leads to hyperactivation of NMDA receptors, which damages hippocampal synapsis and is known to disrupt learning and memory formation [46,81]. Blocking D-serine production under inflammatory conditions has been related to improved outcome following brain trauma [81]. 

Sphingomyelin can be spectrally identified by a characteristic set of peaks in the range of 717–720 cm^−1^ and 1295–1299 cm^−1^, due to the symmetric stretching of C-N and the twisting of CH_2_, respectively [114,119] (Figure 4c). Sphingomyelin is a prominent sphingolipid, found in the plasma membrane with a distribution ratio correlating specifically with cholesterol for suitable membrane function [147]. This lipid plays vital roles within the CNS in terms of cellular growth, differentiation, and death [82]. In the brain, sphingomyelin is also associated with improved cognitive maturation, brain myelination, and regulation of inflammatory responses [83,148]. Levels of sphingomyelin have been shown to increase following TBIs for over 12 months, rendering it a neuro-marker for both sub-acute as well as chronic phases [84]. Disruption of sphingomyelin metabolism is known to dysregulate the mitochondrial energy pathways, slowing the healing process and increasing the risk of secondary injuries, contributing to the overall TBI pathogenesis [149]. 

Sulfatides are a class of sulfo-lipids, abundantly present in brain tissue and are a main feature in myelin sheath and within myelinating cells [87]. Spectral characteristic fingerprints of sulfatides are shown in Figure 4d. These representative peaks include the stretching of the C-OH from the sulphated pyranose ring and C-C backbone along with the CH_2_ wagging in the range of 888–893 cm^−1^ [128,129,130], with the anti-symmetric stretching of the SO_4_ group identified at 1107–1111 cm^−1^ [150]. Sulfatides, directly derived from galactocerebrosides, constitute promising candidate TBI biomarkers due to their roles in protein trafficking, neuronal transduction, and negative regulation in oligodendrocyte differentiation [151]. Following TBI, sulfatide levels decrease in the brain subsequently, allowing the oligodendrocyte production to occur at a higher rate, thus providing axonal myelin wrapping with improved action potential and transmission. However, due to the functional role of sulfatides within the membrane, the dysregulation can lead to physiological responses such as seizures [87]. 

**Figure 3 cells-12-02589-f003:**
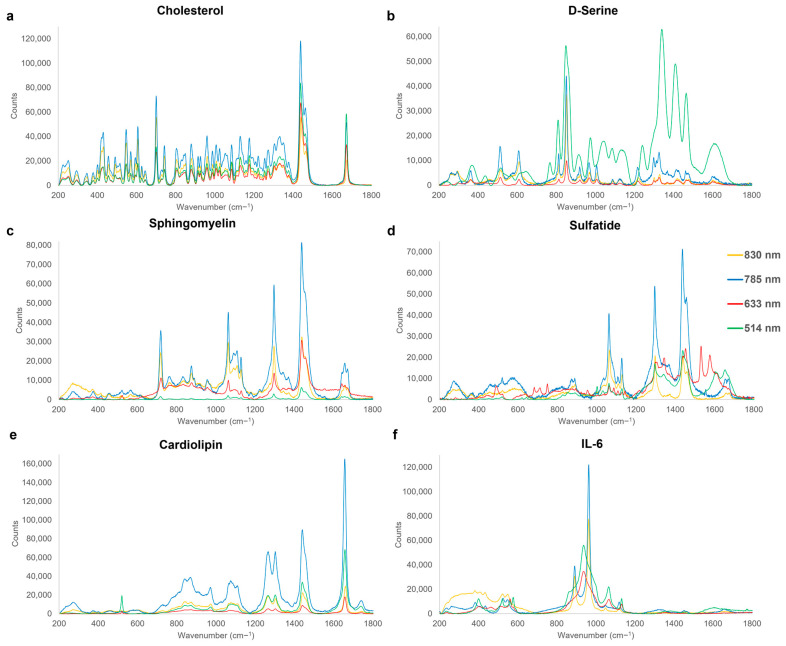
Characteristic Raman spectra fingerprints of sub-acute-phase TBI biomarkers, acquired at excitation wavelengths of 514 nm (green), 633 nm (red), 785 nm (blue), and 830 nm (yellow). Significant Raman peak assignments for each are summarised in Table 3.

Cardiolipin, a mitochondrial-specific phospholipid, which signals damaged mitochondria [88] (Figure 4e), is found to exhibit dominant PO_4_ vibrations at 959–964 cm^−1^ and C-O stretching from the linoleic acid groups in the range of 1266–1268 cm^−1^ [152,153]. Cardiolipin is known to be central to mitochondrial function, with roles in both regulation of metabolism and membrane structure maintenance [88], with damaged and dysregulated mitochondria triggering neuronal death [89,90]. Cardiolipin is externalised from the inner mitochondrial membrane to signal damaged mitochondria and promote mitophagy, ultimately limiting further neuronal damage and retaining cognitive functions [90], resulting in increases in its concentration linked to the extent of mitochondrial damage [89,90]. 

IL-6, an inflammatory cytokine, yields a distinctive spectral signature (Figure 4f) with peptide bonds bending within the protein detected in the 520–525 cm^−1^ region [112], and a sharp peak at 959–964 cm^−1^, associated with the symmetric stretching of the C-N-C bonds of proteins [112]. The pleiotropic IL-6 is involved in immunity, regeneration, and neural functions [93]. Whilst in healthy brain tissue it is not typically detectable, it has been found to be present within an hour of brain injury, making it a reliable marker of TBI-induced damage [91,92]. IL-6 is further upregulated during inflammatory cascades to help healing including the salvation of neurons; however, a prolonged detection of this biomarker is indicative of sustained inflammation, resulting in more damage over healing [90].

**Table 3 cells-12-02589-t003:** Peak assignments of significant peaks present in the sub-acute TBI biomarker phase. *v* = stretching, *δ* = bending, *τ* = twisting, *ρ* = rocking, *ω* = wagging, *s* = symmetric, *a* = anti-symmetric, *arom* = aromatic, *skel* = skeletal.

Wavenumber (cm^−1^)	Assignment	Origin	References
512–516	*ρ*(COO^−^)	D-Serine	[140]
520–525	*δ*(N-C=O)	IL-6	[112]
549	*δ*(N-C-S)	IL-6	[112]
561	-OH *out of plane deformation*	IL-6	[112]
700–702	*In-plane deformation of B ring*	Cholesterol	[143]
717–720	*v_s_*(C-N)	Sphingomyelin	[119]
809–815	*v*(C-C)	D-Serine	[112]
850–854	*ρ*(CH_2_)	D-Serine	[140]
888–893	*v_arom_*(C-O)_4_, *v*(C-C), *ω*(CH_2_)	Sulfatide	[128,129,130]
917–923	*v*(C-C)	D-Serine	[140]
959–964	PO_4_ *vibration*, *v_s_*(C-N-C), *δ*(C-H)	Cholesterol, IL-6, Cardiolipin	[112,152]
970–974	*v*(C-N)	D-Serine	[140]
1062–1067	*v*(C-O), *v*(C-C), *τ*(NH_2_)	Sphingomyelin, Sulfatide, IL-6	[119,124,134]
1107–1111	*v_a_*(SO_4_), *v*(C-N), *v*(C-C), *v*(C-OH),	Sulfatide, Cardiolipin	[23,150]
1127–1130	*v*(C-N), *v*(C-C), *v_skel_*(C-C)	Sphingomyelin, Sulfatide, IL-6	[135,136]
1176–1179	*v_skel_*(C-C)	Cholesterol	[144]
1266–1268	*v*(C –O)	Cardiolipin	[153]
1295–1299	*τ*(CH_2_)	Sphingomyelin, Sulfatide	[114]
1302–1304	*τ*(CH_2_)	Cardiolipin	[143]
1327–1336	*ω*(CH_2_), *τ*(CH_2_)	Cholesterol, D-Serine	[119]
1436–1441	*δ*(CH_2_), *τ*(CH_2_), *v*(CH_2_/CH_3_)	Cholesterol, Sphingomyelin, Sulfatide, Cardiolipin	[113,154]
1656–1660	*v*(C=O), *v*(C=C)	Sphingomyelin, Cardiolipin	[118,155]
1672–1674	*v*(C=C)	Cholesterol	[144]

### 3.3. Chronic Phase

Chronic TBI biomarkers span a larger timeframe and are particularly important in cases of undiagnosed mTBI, frequently causing patients long-term effects. Certain chronic biomarkers are expressed in the early phase, during the acute phase, and continue to be expressed in the longer-term [40,53,56]. The chosen representative chronic biomarkers (Figure 1) provide important underpinning spectral fingerprints, contributing towards ongoing investigations on chronic biomarkers and their post-TBI mechanisms and, once confirmed, the emerging findings would enable the detected biomarker levels at late time points to be used to identify TBI survivors who are at high risk of progressive neurological damage, triggered by their initial TBI. Rapid detection and an in-depth understanding of chronic TBI biomarkers would aid the development of TBI therapeutics and enable in situ tracking of TBI pathology and injury evolution at hospital. Real-time spectroscopy for TBI monitoring would further improve target management and understanding of injury heterogeneity, evolution, penetrance of pharmacological agents, identify novel targets for intervention, and could lead to the development of modalities for tactful therapeutic modifying therapies in TBIs. Representative average spectra of chronic, TBI, solute form biomarkers are shown in Figure 4 with the corresponding peak assignments summarised in Table 4. 

S100B is a calcium-binding protein expressed in astrocytes and is the first brain biomarker to be recognised and utilised within clinical practice guidelines [15,156]. S100B, known to be neuroprotective and neurotrophic, is regarded as a suitable marker due to its involvement in the modulation of long-term synaptic plasticity and energy metabolism by maintaining calcium concentration and ensuring correct mitochondrial function [94,95]. Characteristic dominant spectral peaks are identified for S100B throughout the entire 200–1800 cm^−1^ region (Figure 4a and Table 4). The most intense bands are at 936–937 cm^−1^, assigned to stretching of the C-C bond of proline and valine [157], and at 1002–1004 cm^−1^, due to aromatic phenyl breathing of the C-C bond of the phenylalanine [119,158]. Post TBI, increased S100B levels accompany structural damage and cellular death [95]. This would result in higher intensity of identified Raman peaks, posing it as a promising marker for Raman-based neuro-diagnostics. Physiologically, the implication of increased S100B levels disrupts the correct calcium homeostasis, leading to mitochondrial failures and damaging long-term hippocampal potentiation [94,95]. 

**Figure 4 cells-12-02589-f004:**
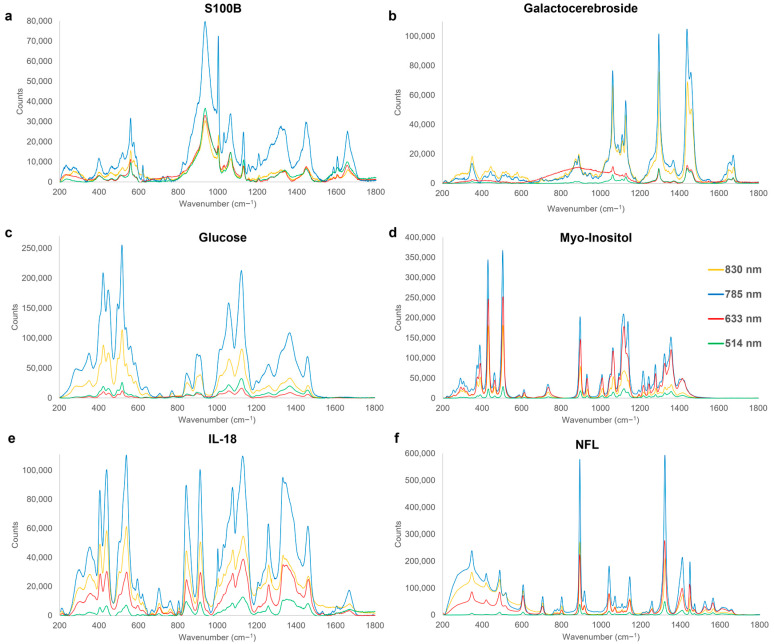
Characteristic Raman spectra fingerprints of chronic-phase TBI biomarkers, acquired at excitation wavelengths of 514 nm (green), 633 nm (red), 785 nm (blue), and 830 nm (yellow). Significant Raman peak assignments for each are summarised in Table 4.

Galactocerebrosides are a major glycolipid of the myelin within the CNS and the most abundant glycolipid component of myelin [159,160]. Figure 4b exhibits dominant spectral peaks in the fingerprint spectrum, particularly, above the 1000 cm^−1^ region. A characteristic peak in the 1128–1133 cm^−1^ range is assigned to the C-N stretching of the amine bond in the acyl chain [119], and the C-C bond skeletal stretching of the acyl backbone common to lipids [136], and further in the 1295–1298 cm^−1^ region, assigned to the twisting of CH_2_ bonds of lipids [114]. This biomarker accumulates in the brain in globoid cells when there is galactosylcerimidase dysfunction, preventing galactocerebroside from degrading, potentially leading to a long-term white matter disease (Table 1).

Overall, peaks in the range of 1295–1299 cm^−1^ are also present in the spectra of sphingomyelin, sulfatide and ganglioside, creating a potential band of interest for the detection of chronic TBI biomarkers via Raman spectroscopy as well as to assess post-TBI recovery and response to interventions. 

Glucose is the predominant and preferred energy source of the mammalian brain, with levels present indicating metabolism in astrocytes [56,100]. Overall, biomarkers in the chronic phase had a larger number of characteristic peaks identified in the lower wavenumber and fingerprint regions 200–800 cm^−1^ than the earlier phases with two intense bands determined from the glucose spectrum (Figure 4c) in the 422–424 and 517–520 cm^−1^ regions, with the former attributed to the in-plane aromatic bending of the C-OH bond [112] and the latter assigned to C-OH bond deformation (Table 4) [112]. Decreases in glucose levels post-TBI have been shown to be associated with poor outcomes due to the indication of an increased need for energy for tissue repair. Glucose metabolism post-trauma has also been shown to lead to an increase in anaerobic glycolysis; however, this has been specifically proven to be a result of ischemia. Furthermore, glucose levels can be compared to other metabolites downstream in the glycolytic pathway such as the ratio of pyruvate (aerobic product) to lactate (anaerobic product). A larger ratio typically correlates to more glycolytic than mitochondrial activity and is associated with worse outcomes, due to an indication that neurons might be too damaged to take up lactate for the TCA cycle [100].

Myo-inositol, a sugar derivative, has been reported to increase in proportion with TBI severity [161], and in correlation with glial proliferation, membrane turnover, and myelin sheath damage (Table 1). Spectral peaks of myo-inositol are highly consistent across the four excitation wavelengths used (Figure 4d) with the most intense bands at 504–506 cm^−1^, assigned to out-of-plane deformation of -OH bonds and further in 1116–1121 cm^−1^ and 1217–1218 cm^−1^ regions, attributed to stretching of the C-C bonds [162,163,164]. Further intense peaks at 896–899 cm^−1^ and 1357–1359 cm^−1^ arise from CH bending within the aliphatic structure of myo-inositol [112]. Increases in myo-inositol levels post TBI provide a good marker of proliferation of glial cells, membrane turnover, myelin sheath damage as well as astrogliosis and cellular osmotic dysregulation [101,102,103,104].

**Table 4 cells-12-02589-t004:** Peak assignments of significant peaks present in the chronic TBI biomarkers phase. *v* = stretching, *δ* = bending, *τ* = twisting, *ρ* = rocking, *ω* = wagging, *s* = symmetric, *a* = anti-symmetric, *arom* = aromatic, *skel* = skeletal, Δ = ring breathing vibration.

Wavenumber (cm^−1^)	Assignment	Origin	Reference
422–424	*In-plane δ_arom_*(C-OH)	Glucose	[112]
504–506	OH *out-of-plane deformation*	Myo-Inositol	[112]
517–520	C-OH *deformation*	Glucose	[112]
559–560	OH *out-of-plane deformation*	S100B	[165]
606–607	CCC *in-plane deformation*	NFL	[166]
804–807	*v_s_* (C-N-C)	IL-18	[112]
842–844	*Out-of-plane δ_arom_*(CH), *ρ* (H(C-O-H))	IL-18	[119,142]
891–893	*v_s_*(COO^−^), *v*(C-C)	NFL	[167,168]
896–899	*v*_aliphatic_(C-H)	Myo-Inositol	[112]
912–914	*v*(C-CH_3_)	IL-18	[169]
936–937	*v*(C-C)	S100B	[157,170]
1002–1004	Δ*_arom_*(C-C)	S100B	[119,171]
1008–1010	Δ*_arom_*(C-C)	Myo-Inositol	[172]
1040–1041	*v*(C-CH_3_), *v*(C-C), *δ*(CH)	NFL	[158,173]
1058–1066	*v*(C-O), *v_skel_*(C-C), *τ*(NH_2_)	S100B, Galactocerebroside, Glucose	[124,134]
1077–1080	*v*(C-C), *v*(C-O)	IL-18	[119,120]
1116–1121	*δ*(CH), *v*(C-C),	Myo-Inositol	[162,174]
1124–1127	*v*(C-C)	Glucose	[135]
1128–1133	*v_skel_*(C-C), *v*(C-N)	S100B, Galactocerebroside, IL-18	[119,135,136]
1143–1146	*v*(C-N), *δ*(CH)	NFL	[138,175,176]
1217–1218	*v*(C-C)	Myo-Inositol	[163]
1259–1261	Amide III (*v*(CN) *δ*(NH)	IL-18	[177,178]
1295–1298	*τ*(CH_2_)	Galactocerebroside	[114]
1321–1323	*δ*(CH), *τ*(CH_2_),	NFL (glycine)	[179,180]
1357–1359	*v*_aliphatic_(CH)	Myo-Inositol	[112]
1368–1371	*δ*(CH_2_)	Glucose	[181]
1409–1411	*v*(C-C), *δ*(C-H)	NFL	[119]
1439–1441	*v*(CH_2_/CH_3_)	Galactocerebroside	[154]
1458–1464	*ω*(C-H), *δ*(CH_3_), *v*(C=O), *v_arom_*(C=C)	Galactocerebroside, Glucose, IL-18	[182,183,184]
1603–1608	*v_a_*(COO^−^), *v*(C=C), *v*(C=O)	S100B	[119,138,185]
1672–1676	*v*(C=C)	Galactocerebroside	[144]

IL-18 is a pro-inflammatory cytokine, produced by microglia and stored in the cytosol as an inactive precursor (Figure 4e) [105]. The most representative fingerprint peaks of IL-18 are identified at 1259–1261 cm^−1^, from the amide III stretching of C-N and bending of N-H, specific to proteins [177,178], and a sharp peak at 806 cm^−1^, assigned to the protein specific symmetrical stretch of C-N-C bonds [112]. Following TBI, IL-18 is known to externalise and bind to the plasma membrane. Uncontrolled or dysregulated IL-18 can lead to releases of neurotoxic enzymes as well as cause chronic inflammatory diseases, whilst IL-18 inhibition has been shown to prevent further damage [106,186].

Finally, the NFL marker has been shown to play a role in maintaining the axonal cytoskeleton [107], and although not necessarily specific to TBI, its rapid increase in CSF is indicative of a considerable axonal injury with change in its concentration correlating the severity of damage [108]. It has been therefore suggested that monitoring the NFL levels is useful for determining the extent of trauma as well as the response to any therapeutic interventions [109]. This marker exhibits a characteristic spectral fingerprint consisting of sharp peaks (Figure 4f), with the most prominent at 891–893 cm^−1^, due to symmetrical stretching of carboxylic acids and C-C bonds [167,168], and the 1321–1323 cm^−1^, attributed to CH bending and CH_2_ twisting modes [179,180].

Further to the spectral assignments of brain-specific molecular species, it is of importance to consider the identified and selected group of correlative changes in Raman peak ratios with key features of these representing quantifiable biochemical changes. The use of characteristic Raman peak ratios allows for the important information to be easily extracted from spectra (particularly, for multiplex mixtures in complex biological environments such as the body fluid or brain tissue) to gain more information and in conjunction with multivariate analysis methods, the ratio-metric inspection and assignment of Raman spectra can be used to identify spectral changes informed by chemical and spectroscopic knowledge, showing clear systematic trends with identifiable features of biomolecules. With these key Raman peak ratios correlating with structural and quantitative biomarker information, analysis of fingerprint spectra could be performed more accurately than the individual peak changes, which could be often masked in complex biofluids. We have thus utilised dominant peak ratios of the solute state TBI biomarkers, which also allows comparison of their variation across the four excitation wavelength lasers (Figure 5).

**Figure 5 cells-12-02589-f005:**
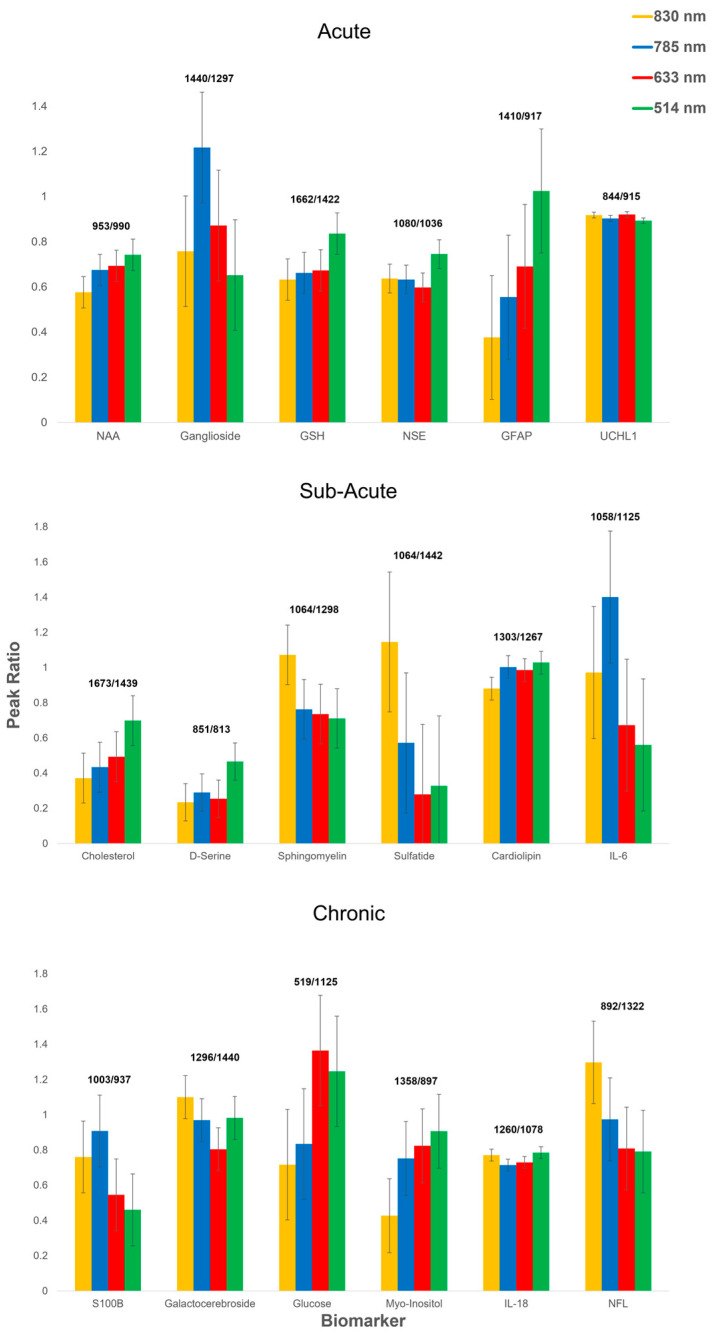
Characteristic peak ratios identified from spectral fingerprints per each studied TBI-indicative biomarker at four excitation wavelengths of 514 nm (green), 633 nm (red), 785 nm (blue), and 830 nm (yellow). (*p* < 0.0001, one-way ANOVA). N-Acetyl-L-aspartic acid (NAA); Glutathione (GSH); Neuron-Specific Enolase (NSE); Glial fibrillary acidic protein (GFAP); Ubiquitin carboxyl-terminal hydrolase isozyme L1 (UCHL1); Interleukin-6 (IL-6); Interleukin-18 (IL-18); and Neurofilament light chain (NFL).

Characteristic peaks were selected without bias from the six assigned bands identified via the peak pick method for each studied biomarker in acute, sub-acute and chronic phases, generating the unique intensity ratios barcode in the fingerprint region. The use of the relative peak ratio intensity provides more accurate and reproducible information over the absolute Raman intensity, which differs among different Raman devices and changing conditions, and is of further importance for multivariate analysis methods often employed in conjunction with Raman spectroscopy detection and analysis [187,188]. These peak ratios yield the combined unique spectroscopic barcode for each brain-injury marker, which could be used as a reference for rapid and accurate diagnostics of TBI in easily accessible biofluids as well as allow to easier quantify the performance of emerging spectroscopic techniques via the standard variation of the ratios between the amplitudes of two Raman bands from a given biomarker over repeated measurements (Figure 5). For the majority of the biomarkers, within the experimental error, there was no significant difference in the identified peak ratios when varying the excitation laser except for GFAP, sulfatide and glucose (*p* *** < 0.0001, one-way ANOVA), which had the largest variance per phase group, with sulfatide having the highest variance of the entire cohort (*σ*^2^ = 0.158). This suggests that greater consideration should be given to the choice of the Raman excitation laser to detect this biomarker, since the molecular bonds of interest are excited at varied intensity strengths depending on the laser wavelength. Conversely, UCHL1, cardiolipin, and IL-18 exhibited the lowest variance within their corresponding phase group, with the lowest overall cohort variance found to be for UCHL1 (*σ*^2^ = 0.000164) with the highest peak agreement amongst the four excitation lasers.

## 4. Current Challenges and Future Outlook

Overall, Raman based diagnosis of TBI pathologies via biofluid detection is emerging as a promising tool for rapid point-of-care diagnostics, and with the addition of the identified molecular profiles, a simple process of a ‘spectroscopic barcode’ will ensure variation due to the presence and/or change in the level of a particular TBI-indicative biomarker and, further, therapeutic treatments and various underlying conditions will be taken into account and reduce the error in the data interpretation of the associated changes, leading towards improved diagnostic accuracy. 

It is worth noting, however, that whilst important progress has been accomplished herein via Raman spectroscopy spectral fingerprints of biomarkers of TBI, several limitations and challenges remain. Firstly, in terms of study design, biomarkers were sourced with the intention to avoid Raman-active additives such as buffers, which would impact the accuracy of the characters’ fingerprint. This has, therefore, removed several candidate biomarkers identified in the literature since pure forms were not available. Furthermore, we used a varied source of human/animal TBI biomarkers, with five out of the overall eighteen investigated biomarkers derived from animals, due to limitations in availability. Whilst animal model systems frequently share similarities of specific aspects of TBIs, providing the opportunity to target specific biomarkers, animal neuromarker models, however, can occasionally recapitulate only parts of the pathology observed in human TBI [189,190,191,192,193,194]. Recognising the opportunities and limitations of translational biomarkers is therefore critical to advancing human TBI therapies. Importantly, this study aims to bridge the gap between preclinical and clinical research as well as to better demonstrate interactions between central expression of TBI pathological markers and peripheral examinations, which could subsequently be used for diagnosis and evaluation of treatment efficacy. Nevertheless, it is important to note, that we did not carry out spectroscopic detection and classification of TBI vs. healthy biomarkers but rather established the characteristic fingerprints for each chosen biomarker, which is a standalone spectroscopic barcode for either human- or animal-derived neuromarkers. In each case, there is a value and significance to having the ‘baseline’ fingerprint data, as in the case of a rapidly evolving and multifaceted pathology such as TBI, they may provide integrative and complementary clinically useful information, or may be useful for different investigations, time points, or settings. For instance, animal biomarker models provide a major advantage for investigating preclinical measures of therapeutic efficacy [189,190]. Faster and less expensive results can be garnered from these allowing only the most promising candidates to advance to more costly clinical trials (e.g., Sabbagh et al. [195]) and represent an important approach to exploring the translational utility of a novel compound. In preclinical systems, animal biomarkers could also be effective for valuing the translational worth of a compound before costly human trials are initiated. Animal model biomarker assessments can verify proof of concept for a candidate treatment. Furthermore, the identification of consistent biomarkers may shed light on core mechanisms responsible for the disorder, as well as interactions between several of the identified pathologies. Notably, the subsequent use of both human and animal biomarkers can form a bridge between animal and human studies to facilitate translational research. By using the same biomarkers in animals and humans, observations in the preclinical stage can be more easily extended to clinical trials. This forward translation can be complemented by a backward one, where observations in humans can be studied via animals.

An additional challenge is that the spectral bands associated with some biomarkers are also present in several other proteins, and so these changes cannot always solely be attributed to a particular molecule. Therefore, a broad panel of biomarkers should be used in such cases, essential to delivering a diagnosis with high accuracy and specificity. This, with further emerging novel hybrid Raman-AI techniques, refs. [196,197], will in turn enable supporting large-scale triaging and screening for early signs of brain injury, particularly useful for identifying high-risk patients and implementing intervention measures, given both the increased incidence and the prevalence of TBIs. Indeed, considerable research in recent years has focused on potential biomarkers to improve diagnosis and patient phenotyping to enable timely and targeted management. For instance, S100B has been implemented in Scandinavian countries and the ALERT-TBI study showed high sensitivity for the combination of GFAP and neuronal UCHL1 measured within 12 h of injury, forming the basis of the first FDA-approved TBI test for triaging the need for CT scanning [198].

One further issue that needs to be addressed for real-world adoption is a lack of standardisation in methods used within the current literature, from matters pertaining to biofluid collection, storage and pre-processing, to those concerning Raman measurement protocols and chemometric analyses and classification. Standardisation of protocols and conducting larger-scale clinical studies are vital for advancing the clinical translation of Raman spectroscopy biomarker profiling in TBI diagnostic and prognostic applications. 

Finally, there has been an increase in the literature surrounding the development of Raman spectroscopy-based technologies for TBI detection, utilising surface-enhanced Raman spectroscopy (SERS), microfluidics, and Raman probes [33,35,36,199,200]. As such new spectroscopic technologies are being engineered and prototyped, it is essential to improve the understanding of brain injury biomarkers and their mechanism in various post-TBI phases, which will continue to grow alongside these, particularly when utilising human biofluids and TBI models.

## 5. Conclusions

A Raman spectroscopy fingerprint library has been generated for a large cohort of TBI-indicative biomarkers. The spectral fingerprint of each studied biomarker generates a unique signature and comparisons of certain values for determined frequencies and their raw and aqueous solution states are determined. These spectra provide the fundamental information necessary to track the biochemical changes induced by the neurological markers categorised according to the different stages post-brain injury, i.e., the acute, sub-acute and chronic phases. These can not only assist in understanding the TBI mechanism and check the presence, persistence or fading of characteristic biomolecules in brain tissue or biofluids following injuries but also establish Raman spectroscopy-aided diagnosis of TBI as a complementary point-of-need technique. The longer-term goal would be to identify biomarkers or distinctive patterns, which would help to rapidly detect, triage, and monitor TBIs by analysing the differences in Raman spectral signatures. 

The most dominant peak ratios for each biomarker have been further identified, highlighting the various advantages amongst which, by using the ratio-metric approach over the absolute Raman intensity of individual bands, is elimination of the need to measure the amplitude of each peak, often necessary for ensuring consistency and reproducibility of the intensity and/or wavenumber position, which can be challenging between each measurement, especially when performed by a non-expert at the point of care. 

Reliable point-of-care diagnostic tools, such as Raman spectroscopy, could improve the specificity of diagnosis and triage of TBI and would greatly aid resource availability in healthcare by reducing the number of unnecessary ED visits for mild TBI and increasing the proportion of patients with moderate/severe TBIs appropriately triaged to attend trauma facilities. With millions of ED visits every year and the high cost per visit without treatment or investigation, even a small reduction will result in massive cost savings. Furthermore, the pathophysiology of injury evolution is multi-faceted, and not understood well enough to provide effective pharmacological targeting to reduce burden or progression. An in-depth understanding of the biochemical injury evolution will allow interventions to be used more effectively and identify novel therapeutic targets to guide tactful treatments. The generated spectral library would further assist in the emerging developments of Raman spectroscopy-based technologies, for rapid acquisition of molecular fingerprints of TBI biochemistry to safely measure proxies for cerebral injury via the body fluids, which will soon provide a tangible path towards non-invasive point-of-care diagnostics for TBI. 

## Data Availability

Data supporting reported results can be provided by the authors upon reasonable request.

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
