# Peer review of "Raman Spectroscopy Spectral Fingerprints of Biomarkers of Traumatic Brain Injury"

_cells, 2023, doi:10.3390/cells12222589_

Round 1
Reviewer 1 Report
Comments and Suggestions for Authors
The authors analyzed 18 TBI-indicative biomarkers with Raman spectroscopy in different post-injury phases.
It is not clear from the methods section what kind of biofluid was used and what kind of patients (male/female, injury severity...) they came from. Blood? CSF? Where did the patient samples come from? Or is this just a review and re-analysis of already existing literature data?
Grammatical errors:
line 108: „Thid could not only improve the treatment of TBI through specific targeting of the damage in contrast to the current methods, which mostly rely on symptomatic relief.
„Thid” to „This”
line 143: „per each biomarker taking into” to „per biomarker taking to”
line 213: „strongest peak for the NSE (Fig.2d) at 875-877cm-1 at all four employed” to „strongest peak for NSE (Fig.2d) at 875-877cm-1 at all four employed”
Comments on the Quality of English Languageline 108: „Thid could not only improve the treatment of TBI through specific targeting of the damage in contrast to the current methods, which mostly rely on symptomatic relief.
„Thid” to „This”
line 143: „per each biomarker taking into” to „per biomarker taking to”
line 213: „strongest peak for the NSE (Fig.2d) at 875-877cm-1 at all four employed” to „strongest peak for NSE (Fig.2d) at 875-877cm-1 at all four employed”
Author Response
Please see the attached response letter.

Reviewer 2 Report
Comments and Suggestions for Authors
TBI is a major public health concern affecting millions of people of all ages worldwide. It is well-established that the diagnosis of TBI is challenging, causing incorrect patient management, avoidable death and disability, long-term neurodegenerative complications and increased costs, and prognosis. Such challenges are even more significant in mild TBI, where missed diagnosis increases the risk of repeated exposure to injury. In this study, the authors aimed to use Raman Spectroscopy-based detection to profile a panel of 18 TBI-indicative biomarkers, including N-acetyl-aspartic acid, Ganglioside, Glutathione, Neuron Specific Enolase, Glial Fibrillary Acidic Protein, Ubiquitin C-terminal Hydrolase L1, Cholesterol, D-Serine, Sphingomyelin, Sulfatides, Cardiolipin, Interleukin-6, S100B, Galactocerebroside, Beta-D-(+)-Glucose, Myoinositol, IL18, neurofilament light chain as well as their aqueous solution, generating a unique spectral reference library to aid the development of rapid, non-destructive and label-free spectroscopy-based neurodiagnostic technologies. These biomolecules can provide additional means of diagnosing TBI and examining the severity of injury.
This is an interesting study which can add to the body of our knowledge on an important topic, but there are some major caveats which need to be addressed before the manuscript is considered further. I provide detailed comments below to help the authors improve their manuscript:
1. The abstract is not balanced. The introduction and methods sections of the abstract are extensive, but the results section needs to be enriched. There are no results mentioned in the abstract and it is not clear what are the findings.
2. Specific quantified results with clear explanations should be mentioned in the abstract.
3. Abstract: Is this a human or animal study? This is not clear.
4. Introduction: Line 39: change “range” to “ranges”
5. Introduction: Some specific signs and symptoms of TBI should be mentioned. For example, the following paper discusses headache associated with TBI: Examining the association between traumatic brain injury and headache. J Integr Neurosci. 2021 Dec 30;20(4):1079-1094. doi: 10.31083/j.jin2004109. PMID: 34997731
6. Introduction: Line 48: Add “as” before “the initial insult”
7. Introduction: Line 57: This is wrong: “The GCS uses the patient’s eye, verbal, and motor responses to enumerate the injury and categorise it as either mild (3-8), moderate (9-12) or severe (13-15)”. GCS is inversely proportional to brain damage, meaning as the GCS decreases, the severity of the brain damage is higher; therefore, a GCS of 3-8 is associated with severe brain damage. This mistake should be corrected.
8. Introduction: Line 61: “ED” make sure all abbreviations are introduced in the text when they appear for the first time.
9. Introduction: Line 68: Change “away from this group to those” to “on”
10. Introduction: Line 74: This sentence is redundant and can be deleted: “In polytrauma and severe TBI, better triaging would enable early intervention and the correct patient journey to the most appropriate destination.”
11. Introduction: Line 75: This sentence can also be inferred from the introduction and does not need to be repeated, so it can be removed: “In the hospital setting, improved assessment and support early-decision making of TBI severity will support timely decisions”. In addition, “supporearly” is not correct.
12. Introduction: Line 83: “of patients” should be deleted.
13. Materials and Methods: Line 116: It is not clear to the reader what is the source of this when it is stated “Selected raw biomarkers were used without purification and tested in both solid and solution states” Is this from animal or human tissue?
14. Materials and Methods: Line 123: “Reconstituted biomarker samples” Again, it is not clear whether this is an animal, human or a mixture of both?
15. Materials and Methods: Line 121: This is not clear and needs to be clarified: “The refined selected panel was determined by which biomarkers were obtainable in their purest form without Raman active additives, which could impact spectral signatures.”
16. The most important caveat of the manuscript is that biomarkers are a mixture of animal and human studies. For example, reference 49 is from animal studies, whereas reference 53 is from human studies. Not only does this obscure the findings, but such a mixture of biomarkers is not even mentioned in the Methods section, which can be misleading.
17. Results: Line 153: “based on literature references to TBIs” The literature is a mixture of animal and human studies.
18. Figure 1: Abbreviations should be provided in the figure footnote.
19. Figure What is the order of biomarkers arranged in the table? There should be a logic. It seems they are based on phase, but for example GSH is before ganglioside, but it appears after.
20. Figure 5: Any statistical analysis was done here? Include the significance level.
21. Table 1: abbreviations should be introduced in the table footnote.
22. Table 1: What is the order of biomarkers arranged in the table? There should be a logic
23. Line 176: References are needed here: “Acute TBI biomarkers have been shown to be applicable in triaging, diagnosing and eliminating the presence of TBI at the earliest stages”
24. Line 213: It seems that “the” was supposed to be deleted and has a strikethrough.
25. Line 217: The reference used here is from spinal cord injury, which is different than traumatic brain injury: “NSE is plausible for monitoring post-TBI changes since it is only expressed in a direct correlation to the degree of damage and translocated from the cytosol to cell surface upon stimulation, acting as a plasminogen receptor”. This is a serious caveat and can be misleading.
26. Line 326: Some chronic biomarkers actually are expressed early (during the acute phase) but they continue to be expressed for long-term. This should be clarified: “Chronic TBI biomarkers span a larger timeframe and are particularly important in cases of undiagnosed mTBI, frequently causing patients long-term effects.”
27. Line 445: What is meant by “greater consideration” in “This suggests that greater consideration should be given to the choice of the Raman excitation laser to detect this biomarker.”
28. A paragraph at the end of the discussion is required to discuss the limitations, including technical and study design limitations.
29. Conclusion: some of the conclusions actually overlap with the discussion.
30. Conclusion: avoid using references in the conclusion, and the conclusion should just provide a summary of the major findings and no new references should be introduced so as not to confuse with the Discussion.
Comments on the Quality of English Language
Some minor corrections are required, as I have explained previously.
Author Response

(The authors gave the same response as above.)

Round 2
Reviewer 2 Report
Comments and Suggestions for Authors
The authors have addressed my comments.